# The association between cognitive ability and body mass index: A sibling-comparison analysis in four longitudinal studies

Liam Wright[1]*, Neil M. Davies[2,3,4,5], David Bann[1]

1 Centre for Longitudinal Studies, Social Research Institute, University College London, London, United Kingdom, 2 Division of Psychiatry, University College London, London, United Kingdom, 3 Department of Statistical Sciences, University College London, London, United Kingdom, 4 Medical Research Council Integrative Epidemiology Unit, University of Bristol, Bristol, United Kingdom, 5 K.G. Jebsen Center for Genetic Epidemiology, Department of Public Health and Nursing, NTNU, Norwegian University of Science and Technology, Trondheim, Norway

* liam.wright@ucl.ac.uk

## Abstract

**Data Availability Statement:** Data are publicly available via the NLS Investigator (https://www.nlsinfo.org/investigator) and Wisconsin Longitudinal Study (https://www.ssc.wisc.edu/wlsresearch/) websites. Code to replicate the

### Background

Body mass index (BMI) and obesity rates have increased sharply since the 1980s. While multiple epidemiologic studies have found that higher adolescent cognitive ability is associated with lower adult BMI, residual and unobserved confounding due to family background may explain these associations. We used a sibling design to test this association accounting for confounding factors shared within households.

### Methods and findings

We used data from four United States general youth population cohort studies: the National Longitudinal Study of Youth 1979 (NLSY-79), the NLSY-79 Children and Young Adult, the NLSY 1997 (NLSY-97), and the Wisconsin Longitudinal Study (WLS); a total of 12,250 siblings from 5,602 households followed from adolescence up to age 62. We used random effects within-between (REWB) and residualized quantile regression (RQR) models to compare between- and within-family estimates of the association between adolescent cognitive ability and adult BMI (20 to 64 years). In REWB models, moving from the 25th to 75th percentile of adolescent cognitive ability was associated with −0.95 kg/m$^2$ (95% CI = −1.21, −0.69) lower BMI between families. Adjusting for family socioeconomic position reduced the association to −0.61 kg/m$^2$ (−0.90, −0.33). However, within families, the association was just −0.06 kg/m$^2$ (−0.35, 0.23). This pattern of results was found across multiple specifications, including analyses conducted in separate cohorts, models examining age-differences in association, and in RQR models examining the association across the distribution of BMI. Limitations include the possibility that within-family estimates are biased due to measurement error of the exposure, confounding via non-shared factors, and carryover effects.

analysis (including NLSY variable tagsets) is available at https://osf.io/b4anm/.

**Funding:** DB is supported by the Economic and Social Research Council (http://www.esrc.ac.uk; ES/M001660/1); DB and LW by the Medical Research Council (http://www.mrc.ac.uk; MR/V002147/1). NMD works in a unit that receives support from the University of Bristol and the UK Medical Research Council (MC_UU_00011/1) and is supported by a Norwegian Research Council Grant number 295989. The funders had no role in study design, data collection and analysis, decision to publish, or preparation of the manuscript.

**Competing interests:** The authors have declared that no competing interests exist.

**Abbreviations:** ASVAB, Armed Services Vocational Aptitude Battery; BMI, body mass index; MR, mendelian randomization; NLSY, National Longitudinal Study of Youth; PIAT, Peabody Individual Achievement Test; REWB, random effects within-between; RQR, residualized quantile regression; SEP, socioeconomic position; WLS, Wisconsin Longitudinal Study.

## Conclusions

The association between high adolescent cognitive ability and low adult BMI was substantially smaller in within-family compared with between-family analysis. The well-replicated associations between cognitive ability and subsequent BMI may largely reflect confounding by family background factors.

## Author summary

### Why was this study done?

- Obesity is a major contributor to global disease burden, and its prevalence is expected to continue to rise.
- While obesity rates have increased, body mass has not increased uniformly across the population, suggesting a role of individual characteristics in determining obesity; one such characteristic is cognitive ability.
- Existing studies reporting links between cognitive ability and obesity have made adjustment for observed confounders only; they may thus be biased by residual or unobserved confounding.

### What did the researchers do and find?

- We used data from four cohorts of siblings ($n = 12,250$) to examine the association between childhood cognitive ability and adult body mass index (BMI) within families.
- This approach can account for unobserved factors that may bias an association between cognitive ability and BMI that are shared between siblings, such as family socioeconomic position.
- When looking within families, we found little evidence of an association between cognitive ability and (lower) BMI, contrary to results of conventional analysis and existing studies in this literature: Moving from the 25th to 75th centile of cognitive ability was associated with a $-0.06 \text{ kg/m}^2$ ($-0.35, 0.23$) difference in BMI.

### What do these findings mean?

- The results suggest that existing findings on the link between cognitive ability and BMI are biased by shared family factors.
- Given that associations between cognitive ability and other health outcomes have been found using similar observational research designs, sibling data may be useful for assessing potential bias for these health outcomes, too.

## Introduction

Obesity rates have increased dramatically across developed countries since the 1980s [1], with significant consequences for public health and national economies [2,3]. Explanations for rising obesity rates have highlighted the "Big-Two," changes to diets—resulting from wider availability of high energy density foods—and reductions in physical activity, though several other factors have also been proposed [4–6]. Despite numerous policy efforts to tackle rising obesity rates [7,8], and increasing public knowledge of its deleterious consequences, obesity remains a leading cause of disability and mortality [3,8].

While the prevalence of obesity (body mass index [BMI] $\geq$ 30 kg/m$^2$) has increased in developed countries, changes across the full distribution of BMI have not been as pronounced, with relatively little change in the prevalence of underweight and relatively small increases in median BMI [1,9–13]. This suggests that individuals differ in their exposure to the causes of obesity or their susceptibility to these causes [14]; estimates of the heritability of BMI from twin studies range 47% to 90% [15]. Thus, while obesity has increased in response to societal change, individual factors still have a role. Identifying these factors is paramount if the causes of the obesity epidemic are to be understood and potential means to address it are to be to be found.

One characteristic that has been proposed to influence BMI—and health, more generally—is cognitive ability. The field of cognitive epidemiology [16] has gathered substantial evidence that higher cognitive ability is associated with better health outcomes, including lower rates of mortality and major chronic diseases [17]. Cognitive ability is thought to affect health for two main reasons [17–19]. First, cognitive ability is related to higher socioeconomic position (SEP) —for instance, greater earnings and higher educational attainment [20]—and cognition may operate indirectly through this factor. In the present setting, by earning larger incomes, individuals with higher cognitive ability are likely to have greater access to healthier and more varied diets and to live in safer, more walkable neighbourhoods. Second, cognitive ability is argued to operate more directly by increasing individuals' ability to understand and use nutritional, and other health-relevant, information. Correctly using nutritional labelling, for example, requires integrating existing nutritional knowledge with literacy and numeracy skills to decode information and make inferences about the healthiness of food items [21]. Health literacy was described in a 2022 WHO report as an "unrecognized obesity determinant" [8], and many people struggle to understand food labels and other health information [22–24]. Importantly, there is considerable overlap between the concepts of literacy and general cognitive ability [18] such that between-person differences in health literacy may be well captured by general intelligence tests [25].

Many studies have examined the association between cognitive ability and BMI [26]. Some have adopted a longitudinal design and investigated the link between childhood or adolescent cognitive ability and adult BMI, given the possibility of reverse causality in cross-sectional data [27]. Existing longitudinal studies span multiple countries, including several Scandinavian countries marked by low social inequality. Most [28–41], but not all [39], find that individuals with higher cognitive ability have lower BMI and obesity rates in adulthood. Effect sizes are typically stronger for obesity than (mean) BMI [33,36,40,41], and there is evidence that associations are stronger at older ages [31,33,37]. The results from longitudinal studies are consistent with mendelian randomization (MR) evidence from samples of unrelated individuals showing an association between genetic predisposition to high (adult) cognitive ability and lower BMI [42].

Though associations between cognitive ability and BMI are widely found, results in observational studies could be explained by unobserved or residual confounding. Specifically, associations may be driven by differences in early family environments, such as early SEP and

parenting practices, that are either unmeasured or difficult to measure with available data (see S1 Fig for a directed acyclic graph): Childhood and adolescent BMI (which are correlated with adult BMI; [43]) are related to maternal cognitive ability [44], and there are strong socioeconomic gradients in BMI and obesity [45,46]. *Dynastic* effects operating via parental genetics to offspring BMI could partly explain genetic associations [42]. Existing studies on the link between cognitive ability and BMI have attempted to capture early SEP by controlling for one or a few high-level variables (e.g., parent's years of education, occupational prestige, and self-reported income), but these have had limited granularity and spanned a narrow age range of childhood or adolescence [28,32,33,47].

Sibling comparison designs offer one approach to mitigate such confounding. On the assumption that early family environments are shared between siblings, comparing outcomes within sibships removes bias arising from family background (including shared dynastic effects). To our knowledge, no studies have used a sibling design to examine associations between early life cognition and BMI during adulthood. In this study, we combined sibling data from four cohort studies to examine the within-family association between adolescent cognition and adult BMI; large samples are required to achieve adequate statistical power in sibling designs [48]. Given existing evidence that associations are stronger for obesity than BMI, we used a novel statistical approach—residualized quantile regression (RQR) [49]—to examine (within-family) associations with cognitive ability across the BMI distribution, rather than just the (conditional) mean.

## Methods

### Participants

We used data from four cohort studies, each following participants from adolescence across adulthood and containing measures of adolescent cognitive ability and adult BMI: the National Longitudinal Surveys of Youth 1979 and 1997 (NLSY-79 and NLSY-97, respectively), the NLSY-79 Child and Young Adults (NLSY-79 C/YA), and the Wisconsin Longitudinal Study (WLS). These cohorts are described in detail in S1 File. Briefly, the NLSY-79 [50] and NLSY-97 [51] are ongoing studies of young people in the United States that began in 1979 and 1997, respectively. Recruitment to the studies took place at the household level, meaning that multiple sibling sets are included. Beside a nationally representative sample, the NLSY-79 and NLSY-97 also include an oversample of ethnic minority (and, for the NLSY-79, economically disadvantaged) individuals, which we treated as separate cohorts. The NLSY-79 C/YA is a study of the children of all females who participated in the NLSY-79. The WLS is a study of graduates from high school in Wisconsin in 1957 (born 1937 to 1938) with a randomly selected sibling joining in 1977 or 1993. In each cohort, we selected sets of *full* siblings, restricting to +/−5-year difference in birth years to improve the plausibility of the assumption of shared family background (in sensitivity analyses, we vary this restriction).

This is a secondary data analysis of publicly available data and did not require ethical approval.

### Measures

**Adult body mass index.** We calculated BMI by dividing weight by squared height (kg/m$^2$). We focused on BMI measured at age 20 to 64 to reduce the risk of bias due to differential mortality rates. To remove the influence of outliers and biologically implausible values, we set height values less than 4.5 feet (1.37 meters) and more than 7 feet (2.13 metres) and BMI values outside the range of 13 to 70 to missing (approximately 0.1% of observations).

Adult height and weight were measured by self-report in each cohort. Adult height and weight were measured on multiple occasions in the NLSY-79, NLSY-97, and NLSY-79 C/YA and only once (1992 to 1994) in the WLS (see S1 File for more detail). In cases where weight but not height was available at a given data collection, we used the last observed value for height, provided it was collected at age 20 or later, given the broad stability of height during adulthood [52].

### Adolescent cognitive ability

Cognitive ability was measured in each cohort using validated tests capturing multiple domains of cognition. In the NLSY-79 and NLSY-97, this was tested using the Armed Services Vocational Aptitude Battery (ASVAB), which participants sat in 1980 (age 15 to 23) and 1998 to 1999 (age 13 to 20), respectively. For both studies, and to ensure comparability with other cohorts, we used age-normed centile scores (range 0 to 100) provided with the data. These combine weighted scores for the mathematical knowledge, arithmetic reasoning, word knowledge, and paragraph comprehension subsections of the ASVAB (for the NLSY-79, this is the 2006 norming exercise). In sensitivity analyses, we converted cognitive ability centiles to z-scores instead.

In the NLSY-79 C/YA, cognitive ability was measured using three subscales from the Peabody Individual Achievement Test (PIAT), administered at ages 5 to 14 years (reading comprehension, reading recognition, and mathematical ability). Age-normed centile scores are available for each test, which we averaged. Given the PIAT was administered on multiple occasions, for age comparability with the NLSY-79 and NLSY-97, we used the last available measurement. Note, children were not eligible to complete the reading comprehension test if they obtained a low score on the reading recognition test, so we averaged the reading recognition and mathematical ability if only these were available [44]. Two- and three-test averaged PIAT scores were highly correlated ($\rho$ = 0.97).

Cognitive ability was measured in the WLS using the Henmon–Nelson test, a group-administered, multiple-choice assessment containing verbal and quantitative items. The Henmon–Nelson test was sat by students in all Wisconsin high schools at varying school grades from the 1930s through the 1960s. We again used centile scores, with norming based on national test takers (range 0 to 100).

We report results from statistical models, such that a one unit change in cognitive ability is equivalent to moving from the 25th to 75th centiles of its distribution.

**Socioeconomic position.** We included a measure of early SEP to examine whether adjusting for this factor generates similar associations between cognitive ability and BMI *between families* as *within families*. For the NLSY-79, we used a composite measure of SEP [53] that has been used in multiple studies in the cognitive epidemiology literature [44,47], including a study examining the association between cognitive ability and BMI [28]. The measure averages z-scores for family income, parental occupational prestige, and mother's and father's years of education, each measured in 1978 or 1979. For the NLSY-79 CYA, we followed a similar approach and averaged z-scores for family income and mother's education [44], using observations closest to a participant's 18th birthday (information on father's education and on occupation is not available). For the NLSY-97, we averaged z-scores for mother's and father's years of education and family income (data on occupation are not available), each of which were measured in 1997. For the WLS, we used a variable extracted from a factor analysis of family income (averaged between 1957 and 1960), parental occupational prestige, and mother's and father's years of education (recorded in 1957) that is supplied with the dataset. Note, in each cohort except the NLSY-79 CYA, the measure of SEP is fixed within a sibling set. More information on the SEP variables can be found in the S1 File.

**Covariates.** Sibling designs do not control for confounding factors that vary among siblings. We included variables for cohort, sex, ethnic group (White, Black, and Hispanic), age at BMI assessment, birth order, and maternal age at birth. Other potential confounding variables were not measured consistently across cohorts, though data on childhood health were available in the NLSY-79 and NLSY-97, which we use in a robustness check (see below). Note, cohort and ethnic group are fixed within sibships and were included to reduce bias in between family associations. See S1 File for more detail on the definitions of the covariates.

## Statistical analysis

Our main analytical approach was to estimate linear random effects "within-between" (REWB) [54] models of the following form, pooling data from the individual cohorts:

$$BMI_{iht} = \beta_{0ih} + \beta_1 \cdot \bar{Cog}_h + \beta_2 \cdot \Delta Cog_{ih} + \beta_K \cdot X_{it} + \varepsilon_i \tag{1}$$

$$\beta_{0iht} = \beta_0 + \mu_{0i} + \mu_{0h}$$

$$\mu_{0i} \sim N(0, \sigma_{0i}^2); \ \mu_{0h} \sim N(0, \sigma_{0h}^2)$$

where $BMI_{iht}$ is BMI for individual $i$ from household $h$ at time $t$; $\beta_{0ih}$ is the intercept, comprising a fixed-effect ($\beta_0$) and random intercepts at the individual ($\mu_{0i}$) and household ($\mu_{0h}$) levels; $\bar{Cog}_h$ is the mean level of cognitive ability in a given household $h$ (the between-family effect); and $\Delta Cog_{ih}$ is the sibling-specific deviation from the mean household cognitive ability (the within-family effect). $X_{it}$ is a vector of control variables, specified above, with age, birth order, and maternal age modelled with natural cubic splines (2 degrees of freedom each) [55] to account for potential nonlinearities in their relationship with BMI [10]. To account for potential cross-cohort differences, we also included interaction terms between cohort and sex, ethnic group, age, and (depending on the model) SEP. We estimated models including and not including adjustment for SEP. Our interest was in the change in the coefficient $\beta_1$ (between-family association between cognitive ability and BMI) following adjustment for SEP and the relative size of the coefficient $\beta_2$ (within family association).

The association between cognitive ability and BMI may differ markedly across the cohorts, so we repeated Model 1 for each cohort separately and excluding a single cohort, in turn (excluding individual random intercepts when analyzing the WLS on its own). We also repeated Model 1 including interactions between mean cognitive ability and cognitive ability deviations ($\Delta Cog_{ih}$) and (linear) age as the previous results suggest a strengthening association across the life course [28,33]. As a robustness check, we ran Model 1 including further control for childhood self-rated health (categories: excellent, very good, good, fair, poor) in the subset of cohorts with this data (NLSY-79 and NLSY-97).

Sibling designs can introduce bias through "carryover effects," where one sibling influences another. Estimates may be attenuated if a sibling's cognitive ability is important not just for their own BMI but their siblings' BMI, too (for instance, through modelling a particular health behaviour such as smoking). To explore this possibility, we repeated Model 1 splitting the set of two sibling households according to whether the older or younger sibling had the higher cognitive ability score. On the assumption that older siblings will be more influential, within-family associations between cognitive ability and BMI should be smaller where the older sibling has the higher cognitive ability.

Next, we examined the association between cognitive ability and BMI across the distribution of BMI using RQR [49]. Unlike standard conditional quantile regression [56], RQR allows for the inclusion of (family) fixed effects in quantile regression models while retaining clear

interpretability [57], a necessity here given we have more than one observation per family. RQR involves two steps. In the first step, the exposure variable (here, cognitive ability) is regressed upon control variables using linear regression from which residuals are calculated (we used the same control variables as in the REWB models). Where control variables have been chosen satisfactorily, these residuals represent exogenous variation in the exposure, which can be used to estimate causal effects. In the second step, the outcome variable (here, BMI) is then regressed upon the residuals using quantile regression to obtain estimates of the effect of the exposure upon the unconditional outcome *distribution*. Bootstrapping is used to calculate standard errors and confidence intervals. Here, we used 500 (cluster-robust) boot-straps and produced quantile regression estimates for each decile (10th, 20th, . . ., 90th centiles) of the BMI distribution. As our measure of cognitive ability was time-invariant, we only used one observation per individual in the RQR procedure. In the main analysis, we used the first observation per individual but also repeated the analysis using participants' last observations. To obtain a comparable RQR estimate of the between-family association, we estimated RQR models using data from a single randomly selected individual in each household, dropping the household fixed effects from the first stage regression.

All analyses were performed in R version 4.1.2 [58]. Complete-case data were used in each analysis, given the difficulty accounting for the complex, multilevel structure of the data in multiple imputation models. Participants were deemed lost to follow-up after the point at which they last provided BMI data. Robust regression was used to absorb the household fixed effects in the RQR first stage regression.

There was no prespecified analysis plan for this observational data analysis study. The main analytical plan was decided before access to the data. The specific implementation of this analysis plan was informed by the data upon inspection once accessed. Multiple sensitivity analyses were subsequently planned to test if the main results identified were robust. A sensitivity analysis examining associations in REWB models using sex-concordant and sex-discordant households separately was carried out following a reviewer comment that results may be biased due to sex differences in self-report measures of height and weight. Results were qualitatively similar to those presented below and are available on request. This study is reported as per Strengthening the Reporting of Observational Studies in Epidemiology (STROBE) guidelines (S1 STROBE Checklist).

## Results

### Descriptive statistics

We identified 20,889 individuals (9,726 households) who were part of sibling sets. Of these, 14,560 individuals (6,665 households) had valid data for adult BMI for two or more siblings. Approximately 16% of individuals were excluded due to missing cognitive ability or covariate data, or lack of discordance in cognitive ability by siblings (*N* = 28). The analytical sample size was therefore 12,250 siblings from 5,602 households (118,355 observations), 59% of the adult sibling sample. The mean number of BMI measurements per individual was 9.7 (SD = 7.4; range 1 to 22), and BMI was observed up to age 62. S1 and S2 Tables detail sample sizes and number of measurement occasions for each cohort.

Descriptive statistics for time-invariant characteristics are displayed in Table 1. S2 Fig shows the distribution of cognitive ability in each cohort. Cognitive ability levels were close to the population mean in the NLSY-79 and NLSY-97 Main samples and the NLSY-79 CYA. The NLSY-79 and NLSY-97 Oversamples were below population means, while cognitive ability levels in the WLS were above population means. While there was less variation in cognitive ability scores within than between families, variability was still substantial; the within-family SD for

**Table 1. Descriptive statistics for time-invariant variables by cognitive ability in four cohort studies.** Mean (SD) and N (%). Cognitive ability group refers to an individual's cognitive ability relative to the average level of cognitive ability in their household (i.e., whether they have above or below average cognitive ability among their siblings). Note, this is why the number of households is the same for below and above average cognitive ability groups—cognitive ability group is defined within each household (i.e., whether a given sibling is above or below average cognitive ability for that household).

| Variable | Group, cognitive ability | NLSY-79 Main | NLSY-79 Oversample | NLSY-79 CYA | NLSY-97 Main | NLSY-97 Oversample | WLS |
|---|---|---|---|---|---|---|---|
| N | Total Sample | 2,556 | 1,755 | 2,809 | 1,873 | 628 | 2,629 |
| | Below Average | 1,276 (49.9%) | 897 (51.1%) | 1,396 (49.7%) | 936 (50%) | 319 (50.8%) | 1,314 (50%) |
| | Above Average | 1,280 (50.1%) | 858 (48.9%) | 1,413 (50.3%) | 937 (50%) | 309 (49.2%) | 1,315 (50%) |
| Observations | Total Sample | 45,842 | 27,332 | 13,954 | 21,332 | 7,266 | 2,629 |
| | Below Average | 22,886 (49.9%) | 14,041 (51.4%) | 6,894 (49.4%) | 10,507 (49.3%) | 3,668 (50.5%) | 1,314 (50%) |
| | Above Average | 22,956 (50.1%) | 13,291 (48.6%) | 7,060 (50.6%) | 10,825 (50.7%) | 3,598 (49.5%) | 1,315 (50%) |
| Households | Total Sample | 1,113 | 750 | 1,246 | 889 | 291 | 1,313 |
| | Below Average | 1,113 (100%) | 750 (100%) | 1,246 (100%) | 889 (100%) | 291 (100%) | 1,313 (100%) |
| | Above Average | 1,113 (100%) | 750 (100%) | 1,246 (100%) | 889 (100%) | 291 (100%) | 1,313 (100%) |
| Cognitive Ability | Total Sample | 50.03 (29.69) | 27.93 (22.85) | 51.27 (24.84) | 50.6 (28.97) | 26.56 (22.23) | 62.67 (25.15) |
| | Below Average | 38.91 (27.48) | 18.59 (17.68) | 41.68 (23.63) | 39.67 (26.45) | 17.4 (17.1) | 51.39 (24.42) |
| | Above Average | 61.11 (27.6) | 37.69 (23.56) | 60.74 (22.24) | 61.52 (27.22) | 36.01 (22.96) | 73.95 (20.35) |
| Female | Total Sample | 1,255 (49.1%) | 833 (47.46%) | 1,410 (50.2%) | 916 (48.91%) | 302 (48.09%) | 1,378 (52.42%) |
| | Below Average | 620 (48.59%) | 426 (47.49%) | 725 (51.93%) | 435 (46.47%) | 146 (45.77%) | 678 (51.6%) |
| | Above Average | 635 (49.61%) | 407 (47.44%) | 685 (48.48%) | 481 (51.33%) | 156 (50.49%) | 700 (53.23%) |
| Male | Total Sample | 1,301 (50.9%) | 922 (52.54%) | 1,399 (49.8%) | 957 (51.09%) | 326 (51.91%) | 1,251 (47.58%) |
| | Below Average | 656 (51.41%) | 471 (52.51%) | 671 (48.07%) | 501 (53.53%) | 173 (54.23%) | 636 (48.4%) |
| | Above Average | 645 (50.39%) | 451 (52.56%) | 728 (51.52%) | 456 (48.67%) | 153 (49.51%) | 615 (46.77%) |
| Birth Order | Total Sample | 2.98 (1.8) | 3.74 (2.33) | 1.94 (0.96) | 1.97 (1) | 2.26 (1.12) | 2.19 (1.31) |
| | Below Average | 3 (1.83) | 3.69 (2.28) | 1.97 (0.94) | 2.07 (0.98) | 2.27 (1.09) | 2.21 (1.29) |
| | Above Average | 2.95 (1.77) | 3.78 (2.37) | 1.9 (0.97) | 1.88 (1) | 2.25 (1.15) | 2.18 (1.32) |
| Maternal Age | Total Sample | 25.82 (5.71) | 25.88 (6.34) | 24.58 (4.9) | 25.45 (4.71) | 24.41 (5.12) | 27.32 (5.16) |
| | Below Average | 25.81 (5.79) | 25.69 (6.34) | 24.68 (4.92) | 25.67 (4.75) | 24.42 (5.07) | 27.28 (5.17) |
| | Above Average | 25.83 (5.63) | 26.08 (6.33) | 24.49 (4.88) | 25.23 (4.67) | 24.39 (5.17) | 27.37 (5.15) |

cognitive ability was between 50% and 70% of the between-family SD in each cohort. Overall, values for sex, age, ethnic group, and birth order were similar among those who had above (family-specific) average cognitive ability and below (family-specific) average cognitive ability (see Tables 1 and S3 and S3 Fig). However, there was some evidence that siblings with above (family-specific) average cognitive ability had better childhood health (S4 Fig).

## Associations between adolescent cognitive ability and adult BMI

In between-family analysis, higher adolescent cognition was associated with lower adult BMI; the difference in BMI moving from the 25th to 75th centile of cognition was $-0.95$ kg/m$^2$ (95% CI = $-1.21$, $-0.69$; Fig 1). Adjusting for SEP attenuated the association to $-0.61$ kg/m$^2$ ($-0.90$, $-0.33$). However, in *within-family* analysis, the association was substantially weaker and confidence intervals crossed the null: a $-0.06$ kg/m$^2$ ($-0.35$, $0.23$) difference, approximately 90% smaller than the adjusted between-family association. A weak within-family association between cognitive ability and BMI was found when conducted in each cohort separately, with confidence intervals overlapping the null in all cohorts (Fig 1). Expressed alternatively, for a person of average adult height (1.68m), a $-0.06$ kg/m$^2$ difference in BMI is equivalent to a 0.17 kg lower weight.

Results examining age differences in the association between cognitive ability and BMI also showed a similar difference in between- and within-family results (Fig 2). In *between-family*

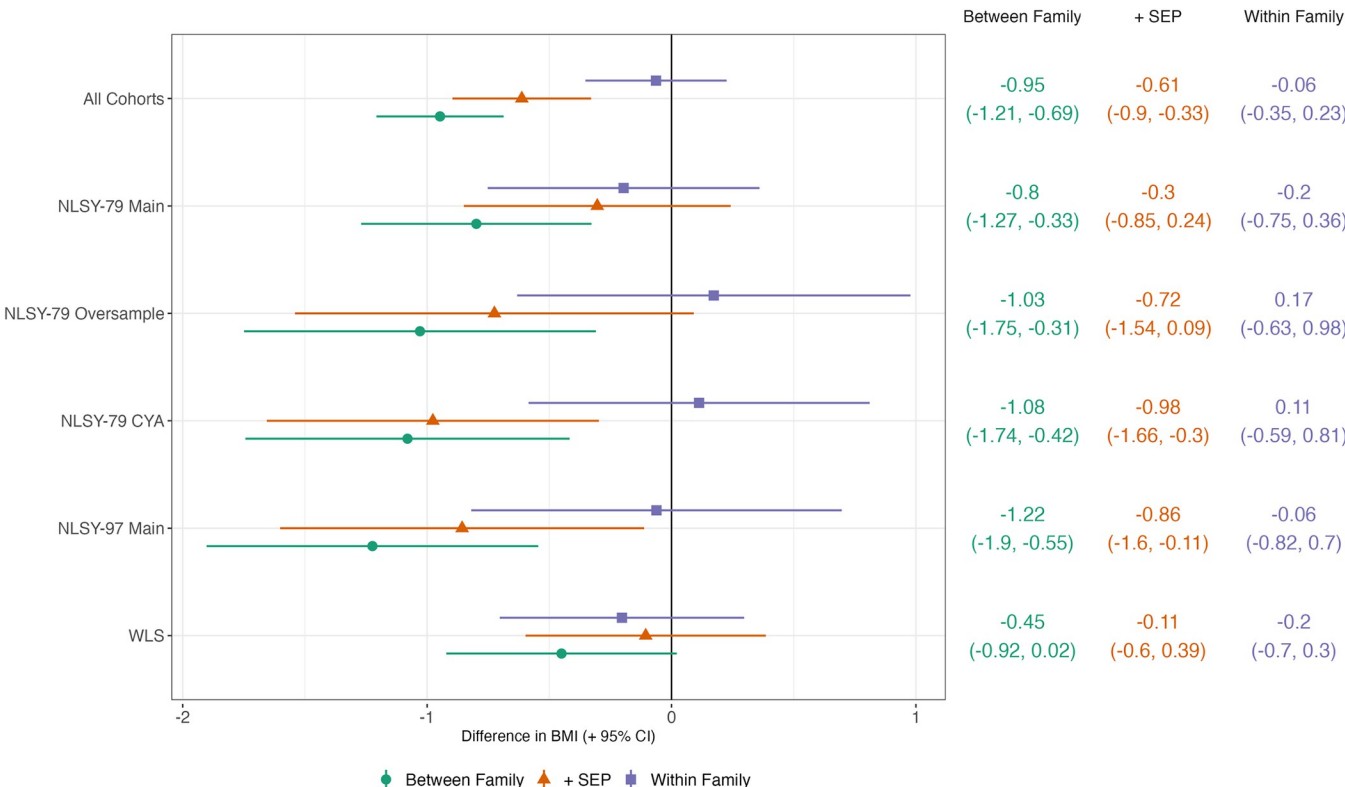

|  | Between Family | + SEP | Within Family |
|---|---|---|---|
| All Cohorts | -0.95 (-1.21, -0.69) | -0.61 (-0.9, -0.33) | -0.06 (-0.35, 0.23) |
| NLSY-79 Main | -0.8 (-1.27, -0.33) | -0.3 (-0.85, 0.24) | -0.2 (-0.75, 0.36) |
| NLSY-79 Oversample | -1.03 (-1.75, -0.31) | -0.72 (-1.54, 0.09) | 0.17 (-0.63, 0.98) |
| NLSY-79 CYA | -1.08 (-1.74, -0.42) | -0.98 (-1.66, -0.3) | 0.11 (-0.59, 0.81) |
| NLSY-97 Main | -1.22 (-1.9, -0.55) | -0.86 (-1.6, -0.11) | -0.06 (-0.82, 0.7) |
| WLS | -0.45 (-0.92, 0.02) | -0.11 (-0.6, 0.39) | -0.2 (-0.7, 0.3) |

**Fig 1. Between- and within-family associations between cognitive ability (centile rank) and BMI (+ 95% CIs).** Estimates show predicted difference in BMI (kg/m$^2$) comparing individuals at the 25th and 75th percentiles of cognition. Derived from linear mixed effects models with random intercepts at the household and individual levels and age (two natural cubic splines), sex, cohort, birth order, and maternal age included as control variables. BMI, body mass index; NLSY, National Longitudinal Study of Youth; SEP, socioeconomic position; WLS, Wisconsin Longitudinal Study.

analysis, the association was substantially stronger at older ages, rising from −0.40 kg/m$^2$ (−0.69, −0.11) at age 20 to −0.96 kg/m$^2$ (−1.25, −0.66) at age 60. The within-family association, however, showed little change with age and was again small with confidence intervals crossing the null (Fig 2).

Finally, analysis of the association between cognitive ability and BMI, across the distribution of BMI, showed a similar pattern of results. In between-family analysis, cognitive ability was associated with lower BMI across most of the BMI distribution (Fig 3). Effect sizes were relatively stronger (in absolute terms) at higher deciles, suggesting lower cognitive ability was associated with a stretching of the BMI distribution—with higher rates of obesity in particular —though confidence intervals overlapped the null. The effect size was 0.0 kg/m$^2$ (−0.35, 0.29) at the 10th percentile and −0.50 kg/m$^2$ (−1.44, 0.37) at the 90th percentile (see S4 Table for full results). In the within-family analysis, cognitive ability was associated with *higher* BMI at lower deciles of the BMI distribution and *lower* BMI at upper deciles. However, effect sizes were weak and typically smaller than between-family associations with confidence intervals consistently crossing the null (90th centile = −0.23 kg/m$^2$; 95% CI = −0.99, 0.47; Fig 3).

## Sensitivity analyses

Qualitatively similar results were observed when varying the maximum age range to select sibling sets between 1 and 10 years (S5 Fig), when excluding a single cohort from REWB models (S6 Fig), and when using cognitive ability z-scores rather than ranks (S7 Fig). Within-family

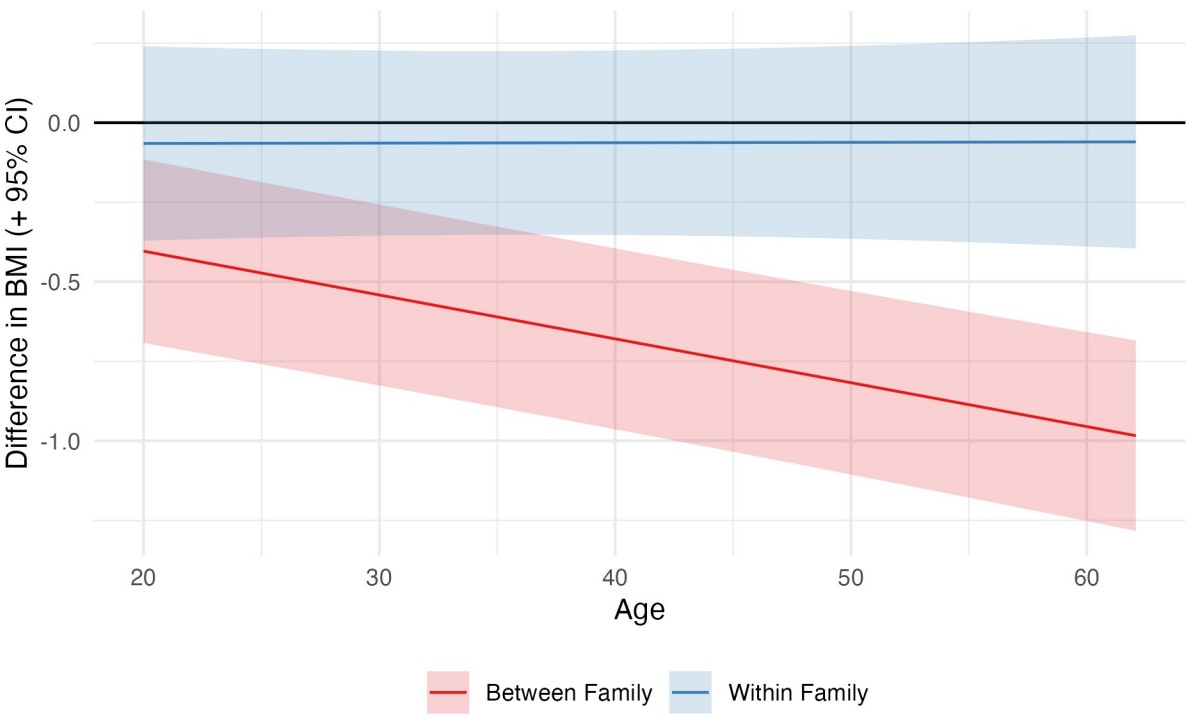

**Fig 2. Between- and within-family associations between cognitive ability (percentile) and BMI by age.** Derived from linear mixed effects models with random intercepts at the household and individual levels and age (two natural cubic splines), sex, cohort, birth order, maternal age, and SEP included as control variables. The lines show, at a given age, the predicted difference in BMI moving from 25th to 75th cognitive ability centiles. BMI, body mass index; SEP, socioeconomic position.

estimates in the NLSY-79 and NLSY-97 combined were attenuated still further when including adolescent self-rated as a control variable. Consistent with a carryover effect, the within-family association was weaker among two sibling families where the older sibling had the higher cognitive ability score, though confidence intervals had considerable overlap (S8 Fig). Within-family associations between cognitive ability and the distribution of BMI generally remained weak when the last observation was used in RQR models (S9 Fig).

## Discussion

### Main findings

Using sibling data from four cohort studies, we found a sizeable difference in the between- and within-family estimates between adolescent cognitive ability and adult BMI. While higher cognitive ability was associated with lower BMI in between-family analysis, effect sizes in within-family analysis were small and confidence intervals overlapped the null. This difference was found across multiple specifications, including analyses conducted in separate cohorts, examining age-differences in association, and examining differences in association across the distribution of BMI.

### Explanation of findings

Our results are consistent with the hypothesis that the association between adolescent cognitive ability and adult BMI observed in this and other studies [28–41] is substantially biased by factors that are shared by siblings. This finding is also consistent with recent work showing a marked attenuation in the genetic correlation between educational attainment and BMI when

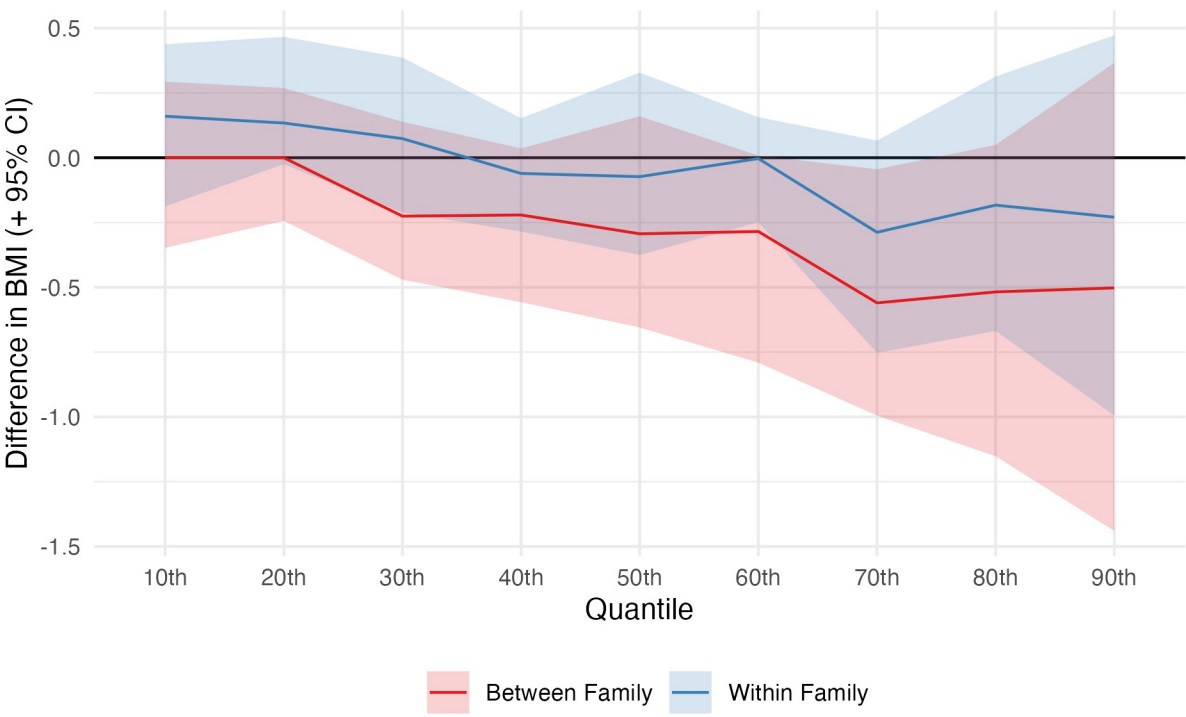

**Fig 3. Associations between cognitive ability (percentile) and BMI by quantile of BMI.** Derived from RQRs using the first observation per individual (within-family effect) and one (first) observation per household (between-family effect). Age, maternal age, birth order, sex, SEP, and cohort were included as control variables in the first stage regressions with household fixed effects also included in within-family models. Confidence intervals calculated using cluster-robust bootstrapping (percentile method, 500 replications). The estimates show the predicted difference in BMI moving from 25th to 75th centile of cognitive ability at a given centile of BMI—i.e., the results for the 50th centile show the predicted difference in the median BMI among persons with 75th centile cognitive ability compared with the median centile of BMI among individuals with 25th centile cognitive ability. BMI, body mass index; RQR, residualized quantile regression; SEP, socioeconomic position.

using sibling data [59]. One plausible confounder that is shared between siblings is childhood SEP, which is, in turn, associated with both adolescent cognition [60] and adult BMI [61,62]. Consistent with this, adjustment for childhood SEP attenuated associations markedly in between-family analysis. However, SEP is a multidimensional construct typically captured with measurement error, leading to possible residual confounding. For instance, years of education does not capture quality of schooling and household income does not capture wealth. Besides SEP, other possible confounders shared between siblings include parenting practices and early neighbourhood (social and built) environments, which may influence the formation of diet and exercise habits, and maternal obesity, which may influence development in utero [63] and, subsequently, offspring BMI into adulthood (at least for extreme maternal obesity) [64]. Further, given that siblings share on average 50% of their genes with other siblings and 50% with each parent, associations could therefore be confounded by shared genetic factors that influence cognition and BMI [65].

While our within-family estimates were still consistent with cognitive ability having a causal effect on BMI, the small effect sizes we identified are surprising given theory and previous results that health literacy and SEP (which cognitive ability should in part operate through) are important predictors of obesity, as well as other results in the wider cognitive epidemiology literature showing little attenuation when using sibling designs for other outcomes [66–68]. We note, however, that at a macro level, policies focusing on teaching the public about food

choices appear to have little evidence of impact [7]. Further, quasi-experimental studies of compulsory schooling reforms—which are associated with small increases in cognitive ability [69]—and natural experiment studies of income windfalls find inconclusive (and sometimes conflicting) effects on obesity overall [70,71]. One reason for the small effect sizes found here may be that non-volitional factors, such as appetite and dietary norms, are of greater importance than conscious, reflective decision-making for eating and physical activity. Though our results suggest a limited causal role for cognitive ability on one's own BMI, it is possible that parental cognitive ability still has importance [44] through influencing family SEP and parenting practices (public health interventions could have indirect effects through parental practices, too).

Another possibility is that our within-family estimates are biased and overly conservative. Sibling designs can introduce bias through three channels: measurement error of the exposure, confounding via non-shared factors, and carryover effects [48,72,73]. While the cognitive tests used here have high reliability [74,75], insofar as adolescent cognitive ability is only proxying for more contemporary cognitive ability levels, changes to cognitive ability over the life course [76] could lead to attenuated associations. This process may be particularly pronounced in within-family analyses given the increasing heritability of cognitive ability (and thus the increasing similarity of siblings) as individuals age [77]. However, we note that we found little evidence of associations between adolescent cognitive ability and adult BMI becoming smaller as individuals aged. Regarding non-shared factors, it is unclear which would bias the association downwards to such an extent—many candidates would be anticipated to upwardly bias the association or are otherwise relatively rare at a population level. In our data, adjusting for childhood self-reported health attenuated estimates still further. Childhood illnesses causing lifelong wasting are unlikely to be sufficiently common. We did, however, find suggestive and indirect evidence of carryover effects—associations were weaker when the older sibling had higher cognition. However, the analysis was low powered and point estimates continued to show relatively small effect sizes. One potential source of carryover effects that was not explored in our analyses was differences in parental investments: There is evidence that parents compensate behaviourally for differences in siblings' polygenic predisposition to high cognitive ability, at least in some families [78].

## Strengths and limitations

Strengths of this study included the use of multiple samples and the longitudinal design: Measures of cognitive ability were collected years before BMI, and we were able to increase statistical precision by including repeat measurement of BMI for each individual. Limitations of this study included a reliance on self-reported height and weight measurements, which likely contain measurement error [79]. Further, the degree of this error could be feasibly related to cognitive ability level. As noted, while sibling designs account for *shared* characteristics, non-shared factors may still bias associations (e.g., childhood infections or disease). There was also substantial attrition in the cohorts, and, by using complete case data, there is a potential for selection bias. However, the number of successful follow-ups was similar among above and below (family) average cognitive ability groups. A final limitation was that our data were from a single country (the US) and that the median year of follow-up was 2004. It is possible that causal effects of cognition on BMI could be present in other study contexts yet absent in the one investigated in this study. For instance, in more recent years, cognitive ability could have become a stronger predictor of SEP or a more important determinant of the behaviors that influence obesity. Future research could test the extent to which associations and causal effects differ by time or place.

## Conclusions

Associations between high adolescent cognitive ability and low adult BMI were partly attenuated by family SEP in between-family analysis and substantially attenuated toward the null in within-family analysis. Our results are consistent with the hypothesis that the well-replicated association between high adolescent cognitive ability and low adult BMI is biased by factors such as SEP, that are shared by siblings. Since such factors may confound associations between cognitive ability and other health outcomes, further research is required to test whether other results in the cognitive epidemiology literature are biased.

## Supporting information

**S1 STROBE Checklist. STROBE checklist.**
(DOCX)

**S1 File. Detailed data description.**
(DOCX)

**S1 Table. Sample selection flow table.**
(DOCX)

**S2 Table. Follow-ups by cohorts and (within-family) cognitive ability group.**
(DOCX)

**S3 Table. Descriptive statistics, time-varying variables.**
(DOCX)

**S4 Table. Residualized quantile regression results.** Analyses using first observation per individual. Between-family effect estimated using one randomly selected individual per household. Age, maternal age, birth order, sex, SEP, ethnic group, and cohort included as control variables in the first stage regressions with household fixed effects also included in within family models. Confidence intervals calculated using cluster-robust bootstrapping (percentile method, 500 replications). Columns headed *excl.* Refer to analyses excluding named cohort and indicate within-family associations.
(DOCX)

**S1 Fig. Directed acyclic graph.** The association between (adolescent) cognitive ability and adult BMI may be confounded by family background (including dynastic genetic effects) and other factors shared between siblings. Existing studies typically attempt to control for family background with measured SEP, which, because of the few high-level variables that are used, can be thought of measuring family background with some measurement error. Thus, controlling for measured SEP does not full block confounding through family background. The sibling design instead accounts for family background by design.
(TIFF)

**S2 Fig. Descriptive statistics and distribution of cognitive ability (percentiles) by cohort.**
(TIFF)

**S3 Fig. Distribution of control variables by (within-family) cognitive ability group.**
(TIFF)

**S4 Fig. Association between (within-family) cognitive ability and probability of poor childhood health (+95% CI).** Estimates drawn from linear probability fixed effects models. Childhood health variables from NLSY-79 and NLSY-97. Self-rated health converted to binary variable for this analysis (poor or fair vs. good, very good, or excellent). Estimates show the

difference in probability of each outcome as cognition centile increases from 25th to 75th centile.
(TIFF)

**S5 Fig. Between- and within-family associations between cognitive ability (centile rank) and BMI by birth year range used to extract sibling sets.** Derived from linear mixed effects models with random intercepts at the household and individual levels and age (two natural cubic splines), sex, cohort, birth order, ethnic group, and maternal age included as control variables.
(TIFF)

**S6 Fig. Between- and within-family associations between cognitive ability (centile rank) and BMI (+ 95% CIs).** Models excluding the named cohort (y-axis). Estimates show predicted difference in BMI ($kg/m^2$) comparing individuals at the 25th to 75th centiles of cognitive ability. Derived from linear mixed effects models with random intercepts at the household and individual levels and age (two natural cubic splines), sex, cohort, birth order, and maternal age included as control variables.
(TIFF)

**S7 Fig. Between- and within-family associations between cognitive ability and BMI by measure of cognitive ability (centile rank or standardized z-score).** Derived from linear mixed effects models with random intercepts at the household and individual levels and age (two natural cubic splines), sex, cohort, birth order, ethnic group, and maternal age included as control variables.
(TIFF)

**S8 Fig. Between- and within-family associations between cognitive ability and BMI in 2 sibling families, according to whether younger or older sibling had higher cognitive ability score.** Derived from linear mixed effects models with random intercepts at the household and individual levels and age (two natural cubic splines), sex, cohort, birth order, ethnic group, and maternal age included as control variables.
(TIFF)

**S9 Fig. Associations between cognitive ability (percentile) and BMI by quantile of BMI.** Derived from RQRs using the first (left panel) or last observation (right panel) per individual. Between-family effect estimated using one randomly selected individual per household. Age, maternal age, birth order, sex, SEP, ethnic group, and cohort included as control variables in the first stage regressions with household fixed effects also included in within-family models. Confidence intervals calculated using cluster-robust bootstrapping (percentile method, 500 replications).
(TIFF)

## Author Contributions

**Conceptualization:** Liam Wright, Neil M. Davies, David Bann.

**Data curation:** Liam Wright.

**Formal analysis:** Liam Wright.

**Funding acquisition:** Neil M. Davies, David Bann.

**Methodology:** Liam Wright.

**Visualization:** Liam Wright.

**Writing – original draft:** Liam Wright.

**Writing – review & editing:** Neil M. Davies, David Bann.

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
