## [Editor Report · Decision Letter 0]

24 Aug 2022

Dear Dr Wright, 

Thank you for submitting your manuscript entitled "The Association Between Cognitive Ability and Body Mass Index: A Sibling-Comparison Analysis in Four Longitudinal Studies" for consideration by PLOS Medicine.

Your manuscript has now been evaluated by the PLOS Medicine editorial staff and I am writing to let you know that we would like to send your submission out for external peer review.

Please re-submit your manuscript within two working days, i.e. by Aug 26 2022 11:59PM.

Kind regards,

Beryne Odeny

Senior Editor

PLOS Medicine

---

## [Decision Letter · Decision Letter 1]

9 Nov 2022

Dear Dr. Wright,

Thank you very much for submitting your manuscript "The Association Between Cognitive Ability and Body Mass Index: A Sibling-Comparison Analysis in Four Longitudinal Studies" (PMEDICINE-D-22-02815R1) for consideration at PLOS Medicine. 

Your paper was evaluated by a senior editor and discussed among all the editors here. It was also sent to independent reviewers, including a statistical reviewer. The reviews are appended at the bottom of this email and any accompanying reviewer attachments can be seen via the link below:

[LINK]

In light of these reviews, I am afraid that we will not be able to accept the manuscript for publication in the journal in its current form, but we would like to consider a revised version that addresses the reviewers' and editors' comments. Obviously we cannot make any decision about publication until we have seen the revised manuscript and your response, and we plan to seek re-review by one or more of the reviewers. 

We hope to receive your revised manuscript by Nov 30 2022 11:59PM. Please email us (plosmedicine@plos.org) if you have any questions or concerns.

We look forward to receiving your revised manuscript. 

Sincerely,

Callam Davidson, 

PLOS Medicine

plosmedicine.org

Please structure your abstract using the PLOS Medicine headings (Background, Methods and Findings, Conclusions).

Abstract Methods and Findings:

* Please include the population and setting, length of follow up, and main outcome measures.

Please include continuous line numbering throughout your manuscript to facilitate further rounds of review.

Please ensure that you cite items in the Supporting Information as described here: https://journals.plos.org/plosmedicine/s/supporting-information

Please define "lost to follow-up" as used in this study. Other reasons for exclusion should be defined.

Please define the length of follow up (eg, in mean, SD, and range) in the main text.

Please ensure the Y-axis scales match between Figures 2 and 3 to facilitate comparison.

Please enlarge the text in Figure 1 as it is currently too small to read.

Please remove the ‘Declaration of interest’, ‘Funding’, ‘Author contributions’, and ‘Data Availability’ Sections from the end of the main text. All of this information is captured as metadata (to be published alongside the article in the event of acceptance) via the submission form questions. 

Please use ‘et al.’ in the references after listing the first six authors (this also applies to the Supporting Information). See https://journals.plos.org/plosmedicine/s/submission-guidelines#loc-references for further guidelines.

Please ensure that the study is reported according to the STROBE guideline, and include the completed STROBE checklist as Supporting Information. Please add the following statement, or similar, to the Methods: "This study is reported as per the Strengthening the Reporting of Observational Studies in Epidemiology (STROBE) guideline (S1 Checklist)."

Did your study have a prospective protocol or analysis plan? Please state this (either way) early in the Methods section.

Comments from the reviewers:

Reviewer #1: Alex McConnachie, Statistical Review

The paper by Wright, Davies, and Bann considers the association between adolescent cognitive ability, and later adult BMI. They demonstrate that these associations are visible only in between-family analyses, and not within-family, suggesting that these associations are probably due to shared confounders within families. This review considers the statistical aspects of the paper.

The authors use a mixed effects model of continuous BMI, and a quantile regression method, suitable for the family data structure. They use multiple cohorts, and consider the sensitivity of the results to each cohort used. My comments are relatively minor.

In the REWB models, the association is reported as the difference in adult BMI between the extremes of the cognitive ability distribution. This is an unnecessarily large difference - for me it would make more sense to report the difference between the 75th and 25th percentiles, i.e. the middle of the upper and lower halves of the distribution.

The abstract uses "points" when "kg/m2" would be more accurate.

In the statistical methods section, age is reported as one of the adjustment variables. I assume this is age as an adult, when BMI is measured? This could be clearer. Also, maternal age and birth order are included in the models. Are these correlated? Within families, these must be quite strongly linked. Is that OK for these models?

I disagree slightly with the interpretation of the quantile regression results. In the between-family analysis, it looks to me as though there is no (or very little) evidence of an association between adolescent cognitive ability and the location of the 10th percentile of the adult BMI distribution, whereas there does seem to be an association with higher percentiles of BMI. I have seen this before. The lower end of the BMI distribution is fairly fixed, but the risk factor (lower adolescent cognitive ability) is associated with a "stretching" of the BMI distribution in adulthood. Not required for this paper, but I wonder if this can be seen in the variance of adult BMI in relation to adolescent cognitive ability?

Reviewer #2: This is a very nicely written sibling analysis of cognitive ability and BMI. The association between higher cognitive ability and lower BMI does not hold when comparing siblings, which suggest family confounding. The data comes from 4 cohort studies, which provides sufficient evidence base. The methods and results are clearly described, and the results are interpreted appropriately. I didn't find anything to improve in the manuscript. 

Reviewer #3: This important paper reports large-scale within-family analysis of associations between cognitive test performance in adolescence and body mass index (BMI) in adulthood using data from about 5000 families in the United States. The key findings is that the association between cognitive performance and BMI is substantially attenuated when comparisons are made between siblings within a family. This attenuation is nearly complete (effect sizes are quite close to zero in the within-family analysis) and much greater than in models that use statistical methods to adjust for socioeconomic differences between families. 

Cognitive test performance and body mass index are negatively correlated. In neurology, this association has been explored as suggesting potential brain-damaging effects of obesity. In life-course epidemiology, the same association has been explored as suggesting effects of cognitive abilities on obesity risk. Overall, there is more evidence to support the latter of these interpretations; in longitudinal studies, differences in cognitive performance tend to precede differences in body mass index. However, the mechanism through which an association between cognitive performance and obesity comes about is unclear. In this article the authors show convincing evidence that differences between the families of high- and low- cognitive performance individuals account for the gradient. 

The sibling comparison design used in this study is a powerful method to separate effects that arise from an individual's own cognitive characteristics from those that arise from family-level factors correlated with those characteristics, including parents' education and income, health behaviors, values, etc. 

My one comment about the paper regards how the authors interpret their data. To my mind, the authors lean too heavily on family socioeconomic differences as the confounder driving cognitive performance associations with BMI. Covariate adjustment for measures of family SES do not account for large fractions of the association between cognitive performance and BMI. I fully agree with the authors that measurement error may lead to incomplete control for SES in the covariate-adjusted between-family models. But the much larger attenuation seen in the within-family analyses suggests that there may be more to the family-level confounding of cognitive performance-BMI associations than socioeconomic status. It would enhance the article to include some discussion of what these other family level factors might be.

Importantly, the authors note that public health messaging around diet and exercise seem to have limited impact on adults' obesity risk. One interpretation of the data reported in this paper is that such messaging may nevertheless impact the way parents raise their children. 

Reviewer #4: This is an interesting study testing between- and within-family associations between adolescent / young adult cognitive ability and BMI measured between ages 20 and 60 years. The study takes advantage of four different cohorts resulting in 2,250 siblings from 5,602 households. The study found in a between-family comparison that higher cognitive ability was associated with lower BMI, and family socioeconomic position attenuated this association slightly. However, in a within-sibling comparison cognitive ability and BMI were not associated, suggesting that the associations are explained by unmeasured familial (shared environmental or genetic) factors.

The authors have used recently developed state of the art statistical analyses to address the study questions. While this is a well-written manuscript, which conveys an important message, there are, however, a few concerns.

Most important one relates to self-reported height and weight. While the authors have discussed this briefly as a study limitation, they have not addressed - in their analyses or discussion - that, according to some studies self-reports of weight and height carry a gender/sex bias, women more often underreport their weight and overreport their hight compared with men.

This may to some extent bias the sibling-comparisons - in sibling pairs discordant for sex. While the models have covaried for sex, I would suggest carrying out stratified sub-group analyses with sex-concordant and sex-discordant sibling pairs. This at least would reduce some of the reporting bias.

Table 1 is very difficult to comprehend, what do below average and above average refer to? The numbers M(SD), N(%) do not make much sense. This needs further clarification. Alternatively, you may want to consider just showing characteristics of your cohorts and total study population without reference to below and above average, would make more sense to me, and would allow more straightforward comparison to other studies.

Please update your literature review. For instance, your literature on statistics of obesity and overweight prevalence and their disease burden is outdated (e.g. 10.1371/journal.pmed.1003198).

Please attenuate your conclusion about family SEP, as it is on the same pathway than cognitive ability; hence, an alternative interpretation of the attenuated associations between cognitive ability and BMI in analyses including SEP is that SEP is a mediator?

[LINK]

---

## [Decision Letter · Decision Letter 2]

10 Feb 2023

Dear Dr. Wright,

Thank you very much for re-submitting your manuscript "The Association Between Cognitive Ability and Body Mass Index: A Sibling-Comparison Analysis in Four Longitudinal Studies" (PMEDICINE-D-22-02815R2) for review by PLOS Medicine.

I have discussed the paper with my colleagues and the academic editor and it was also seen again by two reviewers. I am pleased to say that provided the remaining editorial and production issues are dealt with we are planning to accept the paper for publication in the journal.

[LINK]

We look forward to receiving the revised manuscript by Feb 17 2023 11:59PM.   

Sincerely,

Callam Davidson, 

Associate Editor 

PLOS Medicine

plosmedicine.org

Requests from Editors:

Please include headline numbers in your Author Summary (including sample size and key findings).

Please ensure your STROBE checklist uses section headings and paragraph numbers. Any other identifier (e.g., page or line number) is likely to change during the publication process.

References 3, 12, 19, 42, and 46 contain details of the academic editors which can be removed.

Please include the date accessed for internet sources in your references.

References 49 and 57 are preprints – please confirm these studies are still preprints and, if so, please add [preprint] per our formatting guidelines: https://journals.plos.org/plosmedicine/s/submission-guidelines#loc-references

Please check that the references in your Supporting Information are presented according to the guidance at the link above. 

To help us extend the reach of your research, please provide any Twitter handle(s) that would be appropriate to tag, including your own, your coauthors’, your institution, funder, or lab. Please respond to this email with any handles you wish to be included when we tweet this paper.

Comments from Reviewers:

Reviewer #1: Alex McConnachie, Statistical Review

I thank the authors for their consideration of my original points. I am happy with their responses, and have no further comments to make.

Reviewer #4: The authors have done a good job in answering the concerns that this reviewer had. However, regarding Table 1, which has been clarified, I continue having difficulties in understanding the rows related to households, and why the Ns are constant across total, below average and above average. Perhaps you could add a footnote, which would clarify this detail.

[LINK]

---

## [Editor Report · Decision Letter 3]

21 Feb 2023

Dear Dr Wright, 

On behalf of my colleagues and the Academic Editor, Dr Sanjay Basu, I am pleased to inform you that we have agreed to publish your manuscript "The Association Between Cognitive Ability and Body Mass Index: A Sibling-Comparison Analysis in Four Longitudinal Studies" (PMEDICINE-D-22-02815R3) in PLOS Medicine.

PRESS

Sincerely, 

Callam Davidson 

Associate Editor 

PLOS Medicine